# *Equilibrium flow*: FROM SNAPSHOTS TO DYNAMICS

## ABSTRACT

Scientific data, from cellular snapshots in biology to celestial distributions in cosmology, often consists of static patterns from underlying dynamical systems. These snapshots, while lacking temporal ordering, implicitly encode the processes that preserve them. This work investigates how strongly such a distribution constrains its underlying dynamics and how to recover them. We introduce the *Equilibrium flow* method, a framework that learns continuous dynamics that preserve a given pattern distribution. Our method successfully identifies plausible dynamics for 2-D systems and recovers the signature chaotic behavior of the Lorenz attractor. For high-dimensional Turing patterns from the Gray-Scott model, we develop an efficient, training-free variant that achieves high fidelity to the ground truth, validated both quantitatively and qualitatively. Our analysis reveals the solution space is constrained not only by the data but also by the learning model's inductive biases. This capability extends beyond recovering known systems, enabling a new paradigm of inverse design for Artificial Life. By specifying a target pattern distribution, we can discover the local interaction rules that preserve it, leading to the spontaneous emergence of complex behaviors, such as life-like flocking, attraction, and repulsion patterns, from simple, user-defined snapshots.

## 1 INTRODUCTION

Computer programs are often designed to accept an input, execute a sequence of hidden computations, and return a concise output, discarding most of the intermediate state along the way. Many natural systems work differently. Because they evolve through continuous ordinary-differential-equation (ODE) dynamics, the *process* and the *output* are inseparable: every microscopic configuration is itself a legitimate input / output of the system. When we sample local patterns in space – whether fossil distributions in palaeontological strata or luminosity-color distributions of planets (Russell, 1979; Babusiaux et al., 2018) – we capture static snapshots of an underlying flow. In other words, the observed pattern distribution is almost invariant under the underlying dynamics.

Each pattern distribution therefore encodes at least one minimal set of dynamics that could preserve it. The ability to recover these dynamics is a central challenge in many areas of science and engineering, as it underpins our capacity to understand, predict, and control the behavior of complex systems. For example, in the life sciences, the search for biomedical interventions is shaped by our ability to understand the algorithms and strategies guiding cellular decision-making, while snapshots of behavior are used to infer underlying properties of cognitive agents. Indeed, developmental and evolutionary biology are rife with inverse problems linking genomic information to system-level form and function (Lobo et al., 2014).

Kolmogorov complexity renders a direct search for such minimal dynamics for *patterns* uncomputable (Chaitin, 1975). We instead pose a more tractable inverse problem: *given a distribution of patterns, find dynamical rules that preserve that distribution*. A key empirical clue guides this reformulation. In many physical, biological, and geological systems the macroscopic distribution drifts slowly, whereas microscopic states fluctuate rapidly. Hence, we ask two questions: 1) given a distribution of patterns, how many different dynamics can preserve it? and 2) how close are the potential dynamics to the ground truth?

A rich literature exists on learning dynamics from distributions with low time resolution. The Fokker-Planck-Kolmogorov equation is typically used to model the time evolution of distributions, as demonstrated by works such as (Zhang et al., 2025; Tong et al., 2020; Chardès et al., 2023).

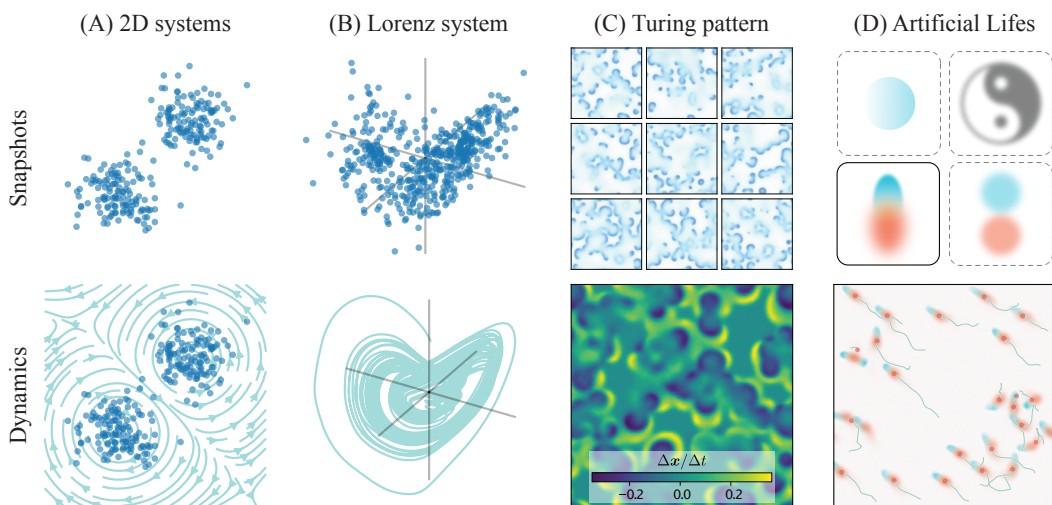

Figure 1: By learning distribution-preserving dynamics, we propose the *Equilibrium flow* method that can infer possible underlying dynamics that could preserve the observed patterns. **(A)** We showcase the application on simple 2D systems. Given a static 2D distribution, we can infer dynamics that do not change the distribution. **(B)** Beyond this, we show the recovered dynamics of the chaotic Lorenz system. The recovered dynamics can preserve the chaotic behavior of the original Lorenz system. **(C)** By applying our method to Turing patterns, we can recover dynamics that preserve both qualitative and quantitative features of the ground truth. **(D)** Furthermore, our method also provides a new way to design artificial life forms. By specifying the target pattern, our method can design local dynamics that preserve the target pattern and generate emergent collective behaviors. The moving trajectories are highlighted with green lines.

However, these approaches either still require some temporal information or rely on computationally expensive neural ODEs to model dynamics and fit distributions via maximum mean discrepancy (MMD) (Gretton et al., 2006) or normalization flows. Some researchers have attempted to address this challenge by adopting spatial grids, but this unfortunately limits applicability to high-dimensional systems (Botvinick-Greenhouse et al., 2023; Yang et al., 2023). Although Li (2023) proposed a method that eliminates the need for temporal information by estimating an explicit pseudo-time for each sample, this approach is limited to periodic systems. Similarly, the work by Arts et al. (2023) directly uses the score function of a density distribution as the force field, but this inevitably misses a significant portion of dynamics that have curl. While distribution-preserving dynamics have been discussed in (Hwang et al., 2005; Rey-Bellet & Spiliopoulos, 2015), these approaches are primarily limited to accelerating sampling procedures for systems with known density functions.

Given these limitations, we propose a new, concise method for learning dynamics from static patterns without requiring temporal information or restricting the form of dynamics. Our approach, which we call *Equilibrium flow*, represents dynamics as a vector field parameterized by a neural network. The training process forces the learned flow to preserve the empirical distribution. Across both simple and complex systems, our method recovers families of dynamics consistent with the data and, crucially, retains qualitative features of the ground truth: reconstructions of chaotic systems remain chaotic, and the reconstructed 2D patterns exhibit similar movement and propagation behaviors (Figure 1).

Our contributions are threefold. Methodologically, we present *Equilibrium flow*, a flexible and broadly applicable framework for inferring dynamics from static data. Theoretically, we use this framework to show that the pattern distribution, along with model biases, imposes strong constraints on the solution space of dynamics. Finally, we demonstrate the generative potential of our method by using it to design novel forms of Artificial life (Langton, 2019).

## 2 LEARNING DYNAMICS FROM STATIC PATTERN DISTRIBUTION

---

**Algorithm 1** Equilibrium flow

---

**Require:** $N$ samples of $d$-dimensional data $X = \{x_i\}$. Number of epochs $T$. Number of trace estimates $k$. Diffusion time $\tau$ (by default, we choose the $\tau$ that equivalent to set $\alpha \approx 0.95$).
 1: $s \leftarrow$ train diffusion model on $X$
 2: $\boldsymbol{v}_\theta \leftarrow$ initialize a neural network with parameter $\theta$
 3: **for** $i \in [1, 2, ..., T]$ **do**
 4:     **for** $\boldsymbol{x} \sim p_\tau(\boldsymbol{x})$ **do**
 5:         $\boldsymbol{v} \leftarrow \boldsymbol{v}_\theta(\boldsymbol{x})$
 6:         $\nabla \cdot \boldsymbol{v} \leftarrow \frac{1}{k} \sum_{j=1}^{k} \boldsymbol{z}_j^\top \nabla_x(\boldsymbol{z}_j^\top \boldsymbol{v}), \quad \boldsymbol{z}_j \sim \mathcal{N}(0, \boldsymbol{I}_d)$         ▷ Hutchinson's Trace Estimator
 7:         $\mathcal{L} \leftarrow \left[\nabla \cdot \boldsymbol{v} + \boldsymbol{v}^\top \boldsymbol{s}(\boldsymbol{x})\right]^2$                        ▷ Minimize equation 1
 8:         Take a gradient descent step on $\mathcal{L}$
 9:     **end for**
10: **end for**

---

To find a continuous dynamics that preserves a given distribution $p(\boldsymbol{x})$, we can consider the probability density as a mass density and map this problem to local mass conservation. By ignoring the noise component, this approach leads us to the continuity equation:

$$\nabla \cdot \left[p(\boldsymbol{x}, t)\boldsymbol{v}(\boldsymbol{x}, t)\right] = -\frac{\partial p(\boldsymbol{x}, t)}{\partial t} = 0. \tag{1}$$

Because $p(\boldsymbol{x})$ is a scalar function, we expand it in this form:

$$p(\boldsymbol{x})\nabla \cdot \boldsymbol{v}(\boldsymbol{x}) + \boldsymbol{v}(\boldsymbol{x})^\top \nabla p(\boldsymbol{x}) = 0. \tag{2}$$

Obtaining the probability density $p(\boldsymbol{x})$ directly is often a challenging problem. To eliminate the $p(\boldsymbol{x})$ term, we note that $\nabla p(\boldsymbol{x}) = p(\boldsymbol{x})\nabla \log p(\boldsymbol{x})$, and obtain:

$$\nabla \cdot [p(\boldsymbol{x})\boldsymbol{v}(\boldsymbol{x})] = p(\boldsymbol{x})\nabla \cdot \boldsymbol{v}(\boldsymbol{x}) + \boldsymbol{v}(\boldsymbol{x})^\top p(\boldsymbol{x})\nabla \log p(\boldsymbol{x}) \tag{3}$$

$$= p(\boldsymbol{x})\left[\nabla \cdot \boldsymbol{v}(\boldsymbol{x}) + \boldsymbol{v}(\boldsymbol{x})^\top \boldsymbol{s}(\boldsymbol{x})\right], \tag{4}$$

where $\boldsymbol{s}(\boldsymbol{x}) = \nabla \log p(\boldsymbol{x})$ is the *score function*, a quantity that can be effectively estimated with pre-trained diffusion models (Ho et al., 2020; Song et al., 2020; Dhariwal & Nichol, 2021) (see Appendix A.3). While the well-known Stein identity (Liu et al., 2016) requires this term to be zero only in expectation, our approach derives a much stronger, local condition by requiring the expression to be zero pointwise for all $\boldsymbol{x}$. This gives us our target equation:

$$\nabla \cdot \boldsymbol{v}(\boldsymbol{x}) + \boldsymbol{v}(\boldsymbol{x})^\top \boldsymbol{s}(\boldsymbol{x}) = 0. \tag{5}$$

The term $\boldsymbol{v}(\boldsymbol{x})^\top \boldsymbol{s}(\boldsymbol{x})$ is straightforward to compute, but the divergence $\nabla \cdot \boldsymbol{v}(\boldsymbol{x})$ is computationally expensive for high-dimensional $\boldsymbol{x}$. We can, however, efficiently approximate it using Hutchinson's Trace Estimator (Hutchinson, 1989):

$$\nabla \cdot \boldsymbol{v}(\boldsymbol{x}) = \mathbb{E}_{\boldsymbol{z} \sim \mathcal{N}(0, \boldsymbol{I}_d)}\left[\boldsymbol{z}^\top \nabla_x\left(\boldsymbol{z}^\top \boldsymbol{v}(\boldsymbol{x})\right)\right]. \tag{6}$$

Thus, equation 1 can be approximated as:

$$\nabla \cdot [p(\boldsymbol{x})\boldsymbol{v}(\boldsymbol{x})] \approx p(\boldsymbol{x})\left[\frac{1}{k}\sum_{j=1}^{k}\boldsymbol{z}_j^\top \nabla_x\left(\boldsymbol{z}_j^\top \boldsymbol{v}(\boldsymbol{x})\right) + \boldsymbol{v}(\boldsymbol{x})^\top \boldsymbol{s}(\boldsymbol{x})\right], \tag{7}$$

where $k$ is a hyper-parameter that determines the number of trace estimations. In this way, we eliminate the need to explicitly compute $p(\boldsymbol{x})$, requiring only the score function $\boldsymbol{s}$, which can be obtained from a pre-trained diffusion model. Since we are locally optimizing $\nabla \cdot [p(\boldsymbol{x})\boldsymbol{v}(\boldsymbol{x})]$ to zero, we can discard the $p(\boldsymbol{x})$ terms and define the training loss function as:

$$\mathcal{L}(\boldsymbol{v}_\theta) = \mathbb{E}_{\boldsymbol{x} \sim p(\boldsymbol{x})}\left[\frac{1}{k}\sum_{j=1}^{k}\boldsymbol{z}_j^\top \nabla_x\left(\boldsymbol{z}_j^\top \boldsymbol{v}_\theta(\boldsymbol{x})\right) + \boldsymbol{v}_\theta(\boldsymbol{x})^\top \boldsymbol{s}(\boldsymbol{x})\right]^2. \tag{8}$$

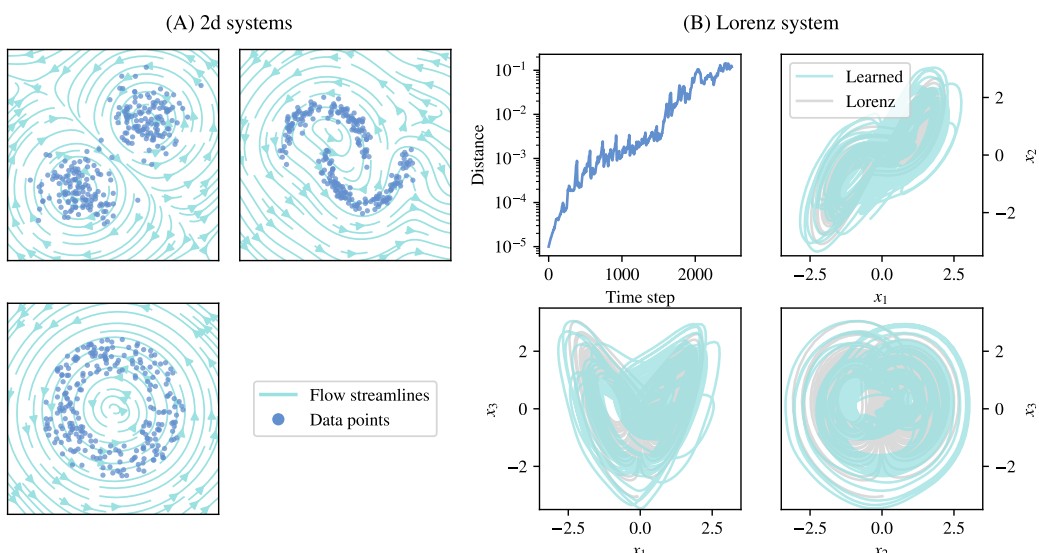

Figure 2: Dynamics learned from static pattern distribution. **(A)** After being trained on samples from statistical 2-D distributions (dots), this method identifies dynamics (arrows) that preserve the distributions. **(B)** After being trained on points sampled from the Lorenz system, the identified dynamics also exhibit chaotic behavior. The top-left figure shows how a small initial difference grows over time, indicating a positive Lyapunov exponent. The other figures compare the evolution of the original Lorenz system with that of the identified dynamics. Although they are not identical, the identified dynamics exhibit a similar overall structure to the Lorenz system and preserve its chaotic behavior.

In practice, diffusion models do not provide the exact score of $p(\boldsymbol{x})$ but rather that of a noise-perturbed (diffused) distribution, $p_\tau(\boldsymbol{x})$, where $\tau \in [0,1]$ represents the noise level. We empirically find that using a small but non-zero noise level strikes an effective balance between training stability and the fidelity of the learned dynamics (see Appendix A.3.1 for details).

Furthermore, a key property of a system with an invariant distribution $p(\boldsymbol{x})$ is that its mean position, $\langle \boldsymbol{x} \rangle = \mathbb{E}_{\boldsymbol{x} \sim p(\boldsymbol{x})}[\boldsymbol{x}]$, must remain constant. This implies that the mean velocity over the distribution must be zero: $\mathrm{d}\langle \boldsymbol{x} \rangle / \mathrm{d}t = \langle \boldsymbol{v}(\boldsymbol{x}) \rangle = 0$. To enforce this physical constraint on our model, we apply a batch normalization layer to the output of the neural network $\boldsymbol{v}_\theta$, forcing it to be zero-mean over each mini-batch. This technique improves training stability and helps prevent the model from converging to trivial solutions. The complete training procedure is detailed in Algorithm 1.

Once the model $\boldsymbol{v}_\theta$ is trained, we can use it to generate continuous dynamics for a given boundary condition. We then apply our method to 2-dimensional data distributions to demonstrate and validate its effectiveness. As shown in Figure 2A, the learned dynamics successfully preserve (or closely approximate) the underlying distribution while enabling individual samples to evolve rapidly. For instance, when the distribution is a mixture of two Gaussian distributions, the learned dynamics is a flow with two vortices in opposite directions. When the distribution is a ring, the learned dynamics is a rotational flow. For more complicated distributions, such as the two-moon distribution in the top-right corner of Figure 2A, the learned dynamics approximately follows the boundary of the distribution.

Subsequently, we extend our approach to samples generated by the Lorenz system. The learned dynamics accurately preserve the original distribution over long time horizons, as demonstrated in Appendix B. Notably, despite having no access to temporal information, our method discovers dynamics that exhibit chaotic behavior – a hallmark of the Lorenz system where trajectories with small initial differences quickly diverge. Although the learned dynamics do not exactly replicate the original Lorenz system, they capture its characteristic shape and fundamental chaotic properties, see Figure 2B. On the top-left figure of Figure 2B, we show how a small initial difference grows over

time, indicating a positive Lyapunov exponent. We applied this method 100 times with different random seeds, and all of them exhibit positive Lyapunov exponents.

## 3 FROM 2D PATTERN TO DYNAMICS

The relationship between static patterns and their underlying dynamics is a fundamental question in science. This problem is often approached from two perspectives. One, rooted in computational complexity theory, quantifies this relationship using measures like Kolmogorov complexity, which is theoretically elegant but generally uncomputable. A complementary line of work, in the pattern-formation literature, seeks to discover the dynamical processes capable of generating an observed pattern. A classic example is Alan Turing's reaction-diffusion model (Turing, 1952), which used differential equations to explain how complex biological patterns can emerge from simple rules. This foundational work inspired subsequent models, such as the Gray-Scott model (Gray & Scott, 1984), that further explore this connection.

A common, often implicit, assumption in pattern formation studies is that an observed pattern strongly constrains its underlying dynamics. Otherwise, a single pattern could be explained by a multitude of theories, limiting their predictive power. This raises a critical question: how many distinct continuous dynamics can generate a given pattern distribution? To investigate this, we apply our method to patterns generated by the Gray-Scott model. This model describes the interaction of two chemical species, $\boldsymbol{u}$ and $\boldsymbol{v}$, through a system of reaction-diffusion equations:

$$\frac{\partial \boldsymbol{u}}{\partial t} = D_u \nabla^2 \boldsymbol{u} - \boldsymbol{u}\boldsymbol{v}^2 + F(1 - \boldsymbol{u}), \quad \frac{\partial \boldsymbol{v}}{\partial t} = D_v \nabla^2 \boldsymbol{v} + \boldsymbol{u}\boldsymbol{v}^2 - (F + k)\boldsymbol{v}, \qquad (9)$$

where $D_u, D_v$ are diffusion coefficients, $F$ is the feed rate, and $k$ is the kill rate. We denote the state of the system as $\boldsymbol{x} = (\boldsymbol{u}, \boldsymbol{v})$ and its dynamics as $\mathrm{d}\boldsymbol{x}/\mathrm{d}t = \boldsymbol{v}_{\mathrm{GS}}(\boldsymbol{x})$. By varying these parameters, the model can produce a wide variety of patterns. Some examples are showing in the first row of Figure 3.

To construct our pattern distributions, we selected four parameter sets that generate distinct life-like (solitons), spiral, wave, and static maze-like patterns (see Appendix C for their parameter settings). For each set, we initiated simulations from random conditions and ran them for $10,000$ steps to ensure the system reached a stationary distribution. The final state of each simulation was then added to our dataset, resulting in $8,192$ pattern samples for each of the four categories. However, applying our original method to such high-dimensional data is challenging due to the computational cost of Hutchinson's trace estimator. To address this, we developed a training-free method that finds a valid subset of solutions using a linear transformation of the score function.

### 3.1 TRAINING-FREE METHOD

For high-dimensional distributions like Turing patterns, training a neural network $\boldsymbol{v}_\theta$ with Hutchinson's trace estimator is computationally prohibitive due to its high memory usage and reliance on second-order derivatives. As an alternative, we propose a training-free approach that can identify a valid subset of solutions. This method constructs a dynamics $\boldsymbol{v}_S$ from a linear transformation of the score function (Hwang et al., 2005):

$$\boldsymbol{v}_S(\boldsymbol{x}) = \boldsymbol{S}\nabla \log p(\boldsymbol{x}), \qquad (10)$$

where $\boldsymbol{S} \in \mathbb{R}^{d \times d}$ is a skew-symmetric matrix (i.e., $\boldsymbol{S}^\top = -\boldsymbol{S}$). As proven in Appendix D.1, $\boldsymbol{v}_S$ is a valid, divergence-free vector field that is orthogonal to the score function. Intuitively, it represents a flow that moves along surfaces of equal probability density.

From the perspective of Helmholtz decomposition, any vector field can be decomposed into a curl-free (gradient) part and a divergence-free (curl) part. Our $\boldsymbol{v}_S$ represents only the curl component. To account for the missing gradient component, which pulls the system towards high-density regions, we add the score function back into the dynamics. This yields an irreversible Langevin SDE (Hwang et al., 2005; Rey-Bellet & Spiliopoulos, 2015):

$$\mathrm{d}\boldsymbol{x} = [\boldsymbol{v}_S(\boldsymbol{x}) + \eta \nabla \log p(\boldsymbol{x})]\,\mathrm{d}t + \sqrt{2\eta}\,\mathrm{d}W_t, \qquad (11)$$

where $\eta$ is a hyperparameter controlling the strength of the gradient term and $\mathrm{d}W_t$ is the Wiener process.

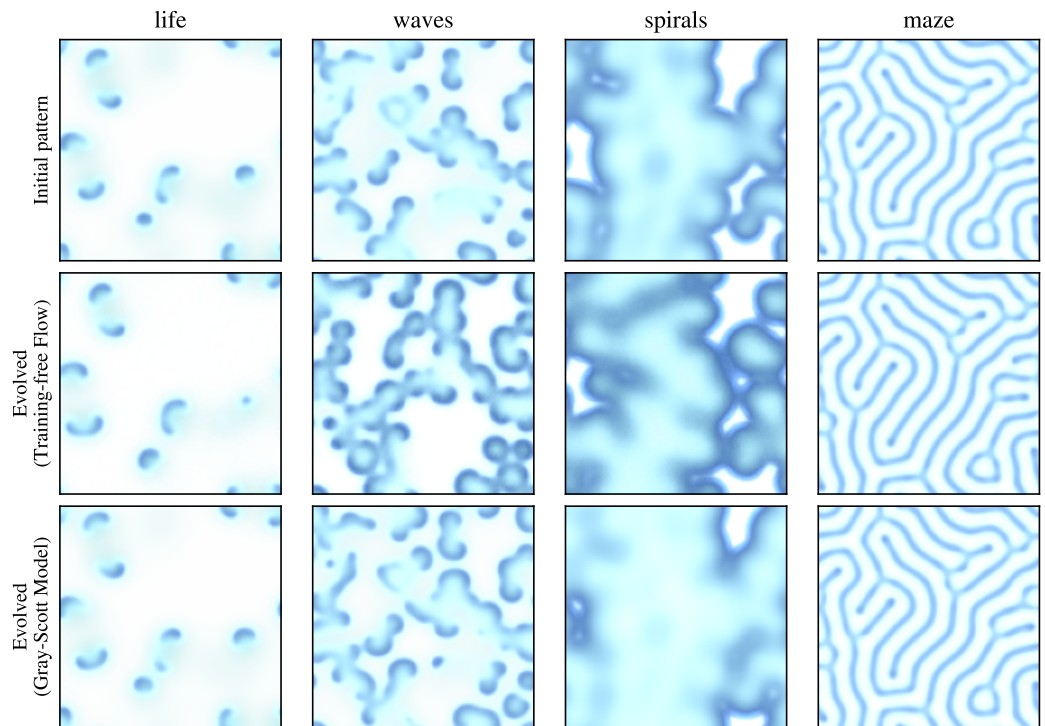

Figure 3: Comparison of dynamics recovered by our training-free method against the ground-truth Gray-Scott dynamics, starting from identical initial conditions. We display four representative pattern types: life-like (solitons), spiral, wave, and static maze-like. Although not identical, the recovered dynamics successfully reproduce key qualitative features of the ground truth, including soliton movement, wave and spiral propagation, and the static nature of maze-like patterns.

For structured data like images, the skew-symmetric transformation $S$ can be efficiently implemented as a convolutional layer with a kernel $K \in \mathbb{R}^{c \times c \times (2r+1) \times (2r+1)}$. The operator is skew-symmetric if the kernel satisfies the condition (see Appendix D.2 for proof):

$$K_{o,i,u,v} = -K_{i,o,-u,-v}, \tag{12}$$

where $o, i$ are channel indices and $u, v$ are spatial indices. For the two-channel Gray-Scott model, we use a simple $1 \times 1$ convolution ($r = 0$) to avoid introducing spatial asymmetries. The kernel $K$ thus becomes a $2 \times 2$ matrix:

$$K_{:,:,0,0} = \gamma \begin{bmatrix} 0 & -1 \\ 1 & 0 \end{bmatrix}, \tag{13}$$

where the scalar $\gamma$ controls the evolution speed and its sign determines the rotational direction of the flow. In our experiments, we test both $\gamma = 1$ and $\gamma = -1$ and select the one whose resulting dynamics are most similar to the ground truth. We set the stabilization coefficient to $\eta = 0.1$ and compute the score with a noise scale of $\tau = 0.1$. The results in Figure 3 show that this approach successfully captures the key qualitative features of the ground-truth dynamics, demonstrating its effectiveness.

## 4 THE UNIQUENESS AND FIDELITY OF LEARNED DYNAMICS

Despite the success of qualitatively recovering complex dynamics (Figures 2 and 3), a fundamental question remains: how unique is the learned solution? That is, how many distinct continuous dynamics can preserve a given pattern distribution, and how closely does a learned solution resemble the ground truth? To investigate this, we conduct an empirical study of uniqueness (consistency among solutions) and fidelity (similarity to ground truth) on two types of systems: the Lorenz attractor and the Turing patterns from the Gray-Scott model.

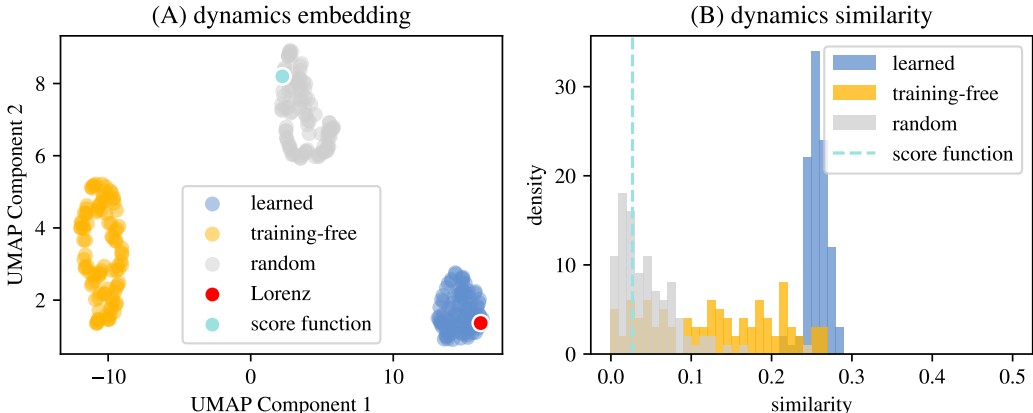

Figure 4: Uniqueness and fidelity of dynamics for the Lorenz system. Representations are formed from vector fields computed at 1024 sample points. **(A)** A 2D UMAP embedding of the dynamics representations. The learned dynamics ($v_\theta$, blue dots) cluster tightly near the ground truth (orange cross), while training-free solutions (green) are more dispersed and random dynamics (gray) are scattered. **(B)** Cosine similarity to the ground-truth Lorenz dynamics. The trained models ($v_\theta$) show the highest and most consistent similarity.

To compare different dynamics, we construct a high-dimensional representation for each one. For a given dynamics $v$, its representation $e_v$ is formed by concatenating its vector field values at $N$ fixed points $\{x_i^*\}_{i=1}^N$ sampled from the data distribution $p(x)$: $e_v = [v(x_1^*) \quad v(x_2^*) \quad \ldots \quad v(x_N^*)]$.

We generate these representations for several dynamics: 100 independently trained models ($v_\theta$), solutions from our training-free method, the ground-truth dynamics, the score function ($s$), and a baseline of 100 randomly initialized networks ($v_R$). Since if $v$ is a solution, so is $-v$, we account for this inherent sign ambiguity by aligning each representation $e_v$ with the ground-truth representation $e_{\text{GT}}$ by ensuring a positive dot product: $e_v \cdot e_{\text{GT}} \geq 0$.

For the Lorenz system, we used $N = 1024$ sample points. We first visualize the space of solutions by applying the UMAP dimension reduction method (McInnes et al., 2018) to the representations, using a cosine metric. As shown in Figure 4A, the 100 learned dynamics ($v_\theta$, blue dots) form a tight cluster near the ground truth, indicating a highly constrained solution space. In contrast, the random dynamics are widely scattered. Figure 4B quantifies this, showing that $v_\theta$ has a much higher cosine similarity to the ground truth than the baselines.

Table 1 provides a detailed quantitative analysis. The learned dynamics $v_\theta$ exhibit exceptionally high self-similarity ($0.99 \pm 0.01$), confirming that our training procedure consistently finds nearly identical solutions. They also achieve the highest fidelity to the Lorenz system ($0.25 \pm 0.01$). Counter-intuitively, the training-free method, which theoretically samples from a constrained set of solutions, shows significantly lower self-similarity ($0.50 \pm 0.29$) and fidelity ($0.13 \pm 0.08$), indicating a wider practical solution space than the neural network approach.

Table 1: Average cosine similarities between dynamics representations for the Lorenz system. The trained models ($v_\theta$) show the highest self-similarity (uniqueness) and the highest similarity to the Lorenz dynamics (fidelity).

| Similarity | Learned ($v_\theta$) | Training-free | Random | Lorenz | Score ($s$) |
|---|---|---|---|---|---|
| Learned ($v_\theta$) | **0.99 ± 0.01** | 0.37 ± 0.20 | 0.08 ± 0.06 | **0.25 ± 0.01** | 0.01 ± 0.004 |
| Training-free | — | 0.50 ± 0.29 | 0.12 ± 0.08 | 0.13 ± 0.08 | 0.02 ± 0.01 |
| Random | — | — | 0.38 ± 0.24 | 0.04 ± 0.04 | 0.09 ± 0.07 |
| Lorenz | — | — | — | 1.00 | 0.017 |
| Score ($s$) | — | — | — | — | 1.00 |

Table 2: Absolute cosine similarity to ground truth for different methods. We compare our training-free method against a modified training-free method with random $K$ rather than skew-symmetric ones, and a re-weighted neural network using $s$ as the dynamics function. Similarities are shown with standard deviations in brackets. Training-free methods do not show standard deviations since there are only two possible $K$ matrices for two channels. Our training-free method achieves the highest similarities across all patterns studied.

|                  | **Life**      | **Waves**     | **Spirals**   | **Maze**      |
|------------------|---------------|---------------|---------------|---------------|
| Training-free    | **0.271**     | **0.589**     | **0.250**     | **0.270**     |
| Random $K$       | 0.021 (0.022) | 0.182 (0.221) | 0.083 (0.090) | 0.072 (0.104) |
| Random Re-weight | 0.052 (0.032) | 0.105 (0.044) | 0.180 (0.106) | 0.031 (0.022) |

This surprising result suggests that the uniqueness observed in the trained models is largely attributable to the *inductive biases* of neural networks, such as smoothness (Rahaman et al., 2019) and a preference for simple functions (Valle-Perez et al., 2019). The training-free method, lacking these biases, may be more sensitive to noise in the score function. The skew-symmetric matrix transformation, which computes weighted differences between state variables, can amplify this noise, especially when dimensions are correlated, leading to a more varied set of solutions.

For the high-dimensional Turing patterns, we assess the fidelity of the dynamics recovered by the training-free method. Due to the stochastic Langevin term in the dynamics, we define the representation as the net change in the pattern after a short evolution from a single initial state $x_0$ to $x_T$: $e_v = \text{flatten}(x_T - x_0)$. The similarities reported in Table 2 show that for each pattern type, the recovered dynamics is most similar to its corresponding ground-truth dynamics compared to random sampled dynamics. Importantly, both the training-free method and random dynamics use the same $\eta = 0.1$ Langevin term, suggesting that the similarity is primarily due to the learned dynamics rather than the score function. This confirms that the method captures pattern-specific dynamics.

Interestingly, these results also support our hypothesis about inductive bias. The training-free method for these patterns used a $1 \times 1$ convolution to implement the skew-symmetric transformation. This structure forces the dynamics at each pixel to be a linear combination of the score function values at that same pixel. Consequently, the recovered dynamics inherits the strong inductive biases of the score function's convolutional architecture, such as locality and translation invariance, which contributes to the successful recovery of structured dynamics.

The added Langevin dynamics also plays a crucial role. Although the learned dynamics may deviate from a specific ground truth, the score function acts as a corrective force, pulling the system back toward the data manifold. This stabilization is particularly important in sparse distributions, where the dynamics are constrained to move within the narrow "tunnels" of the manifold.

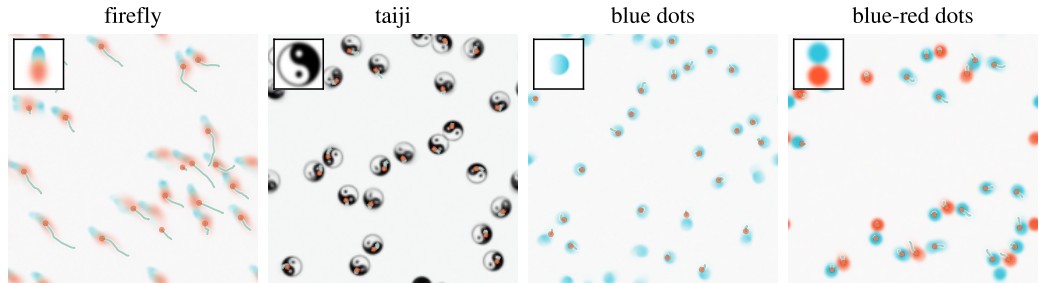

Figure 5: Artificial life forms created via our inverse design approach. We first define a target pattern (top-left corners of each panel) and create a distribution by scattering it with random positions and rotations. Our method then finds the dynamics that preserve this distribution. Trajectories are highlighted with green lines. More complex movements are visualized by using an exponential moving average (EMA). Several systems display emergent properties, such as collective flocking behavior (see "firefly") and pattern-based attraction (see "blue-red dots").

This ability to robustly find dynamics that preserve a given pattern distribution opens up new applications beyond recovering existing systems. It is particularly useful for designing new dynamics, such as *Artificial Life*, a field that studies not only what life is, but *what life could be* (Langton, 2019). Current methods for creating artificial life often rely on discovery through brute-force search (Chan, 2018) or evolution with genetic algorithms (Reinke et al., 2020; Kumar et al., 2025). While powerful, these approaches often require extensive trial and error and offer limited direct control over the final forms and behaviors.

Our method, in contrast, offers a new paradigm for designing such systems through *inverse design*. We can specify a desired pattern, a photograph or a simple hand drawing, and discover the local interaction rules that sustain its distribution. To do this, we scatter instances of the pattern randomly in a 2D space and apply our method to find the dynamics that preserve it. As shown in Figure 5, this process successfully generates artificial life forms that match the desired patterns. The resulting dynamics link a pattern's morphology to its motile behavior. For instance, the highly asymmetric "firefly" pattern leads to smooth movement, while patterns with greater symmetry like "taiji" and "blue dots" have slower motion. Our method thus provides a tool to study the relationship between form and emergent behavior. These systems also exhibit complex *emergent properties* that were not explicitly programmed, such as collective flocking reminiscent of the boids model and attraction or repulsion behaviors.

## 5 DISCUSSION

In this paper, we addressed the fundamental, yet often intractable, problem of inferring process from pattern. Rather than attempting the uncomputable task of finding dynamics for a single static pattern, we posed a more tractable problem: discovering continuous dynamics that preserve an entire distribution of patterns. To this end, we introduced the *Equilibrium flow* method, a framework that successfully recovers plausible dynamics for systems ranging from simple 2D distributions to the chaotic Lorenz attractor and high-dimensional Turing patterns. Our results consistently demonstrate that the learned dynamics exhibit high self-similarity across independent trials and significant fidelity to the ground truth, suggesting the solution space is highly constrained.

Our central finding is that this constraint arises from the combined influence of data sparsity and model inductive bias. High-dimensional patterns from natural systems often lie on low-dimensional manifolds, which inherently restrict the possible trajectories of evolution. The architectural and optimization biases of our learning framework further narrow the space of potential solutions. This dual constraint enables a new paradigm for studying the relationship between patterns and dynamics, with applications in both scientific discovery and creative design. For recovering dynamics, our method opens promising avenues in fields like systems biology and cosmology, where temporal data is scarce but snapshot data is abundant. For designing dynamics, the goal shifts from finding a single ground truth to discovering a set of plausible rules. Our work on Artificial Life exemplifies this, where specifying a target morphology allows us to inverse-design the local interaction rules that sustain it and give rise to emergent collective behaviors.

This work also opens numerous avenues for future research. While we have qualitatively shown that the solution space is constrained, a key theoretical direction is to quantify precisely how properties of the data distribution, such as its sparsity, symmetries, and topology, affect the uniqueness and behavior of the learned dynamics. A deeper understanding here could forge a more tangible, computable link to the problem's abstract framing in terms of Kolmogorov complexity. Methodologically, further investigation is needed to understand the boundaries of our approach and improve its efficiency. The current two-stage process, which relies on a pre-trained score function, could be replaced by more efficient one-stage, end-to-end models. Exploring alternatives to Hutchinson's Trace Estimator or developing ways to deliberately impose beneficial inductive biases on the training-free method are also critical next steps.

In conclusion, we offer a dual contribution. First, we have introduced a novel computational tool for investigating the fundamental relationship between static patterns and the dynamic processes that form them. Second, we provide a practical methodology for inferring temporal evolution from snapshot data, a common challenge across numerous scientific and engineering disciplines. By bridging the gap between static observation and dynamic understanding, we have laid the computational groundwork for a future theory of how process is encoded in pattern.

## 6 REPRODUCIBILITY STATEMENT

We have included the code for our experiments in the supplementary material. To ensure reproducibility, all data for the Turing patterns discussed in Section 3 are generated using programs with fixed random seeds. For the Artificial Life experiments discussed in Section 4, we have included the original PNG files of the target patterns in the supplementary material. The generated Artificial Life videos are also provided in the supplementary material.

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

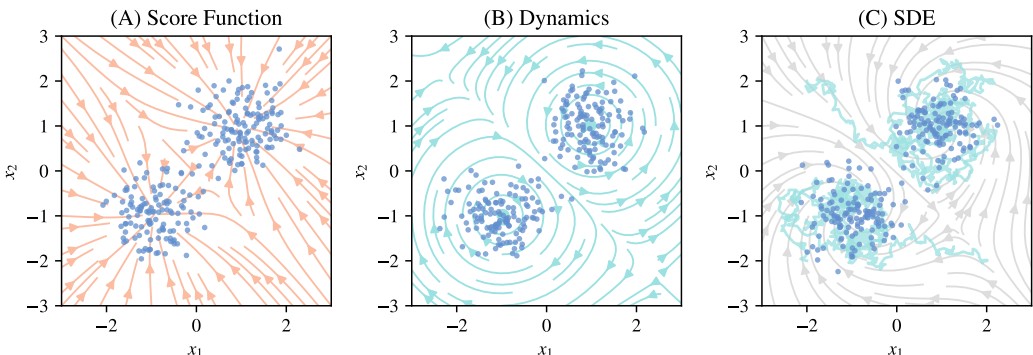

Figure 6: **(A)** The estimated score function (orange streams) trained from sampled data (blue dots) using diffusion model (with $\tau = 0.2$). **(B)** The learned dynamics (blue streams) inferred from the estimated score function and sampled data (blue dots). **(C)** The learned dynamics can be combined with the score function to generate stochastic dynamics ($\eta = 0.25$), shown as blue lines.

# A   NEURAL NETWORK ARCHITECTURE

In this section of appendix, we describe the neural network architectures used in our experiments. We first outline the structure of our dynamic model, which infers the dynamcs from statical patterns, followed by details of the score function model that estimates the gradient of the log probability density.

## A.1   DYNAMIC MODEL FOR 2-D AND LORENZ SYSTEMS

We parameterize the dynamics as ordinary differential equations (ODEs):

$$\frac{d\boldsymbol{x}}{dt} = \boldsymbol{v}_\theta(\boldsymbol{x}). \tag{14}$$

where $\boldsymbol{v}_\theta$ is implemented using a multilayer perceptron (MLP) neural network that maps $d$-dimensional input vectors to $d$-dimensional output vectors.

To enhance the network's ability to capture high-frequency dynamics, we incorporate positional encoding:

$$\boldsymbol{v}_\theta(\boldsymbol{x}) = f_\theta(\boldsymbol{x}, \gamma^n(\boldsymbol{x})), \tag{15}$$

where the positional encoding function $\gamma^n$ ($n = 4$ by default) from (Mildenhall et al., 2021) expands the input dimensions as follows:

$$\gamma^n(\boldsymbol{x}) = \left[\sin(2^0\boldsymbol{x}), \cos(2^0\boldsymbol{x}), \sin(2^1\boldsymbol{x}), \cos(2^1\boldsymbol{x}), \ldots, \sin(2^n\boldsymbol{x}), \cos(2^n\boldsymbol{x})\right]. \tag{16}$$

For the activation function in $f_\theta$, we use SiLU (Ramachandran et al., 2017; Elfwing et al., 2018) instead of ReLU. This choice is critical because our gradient descent process involves vector-Jacobian products that require non-zero second-order gradients (Chen et al., 2019).

## A.2   SCORE FUNCTION FOR GRAY-SCOTT MODELS

We adopt a minimal 2D U-Net, based on the `diffusers` library's `UNet2DModel`, to learn the score function for the Turing patterns. The network receives a two-channel $128 \times 128$ grid, representing the chemical concentrations, and outputs a two-channel grid of the same size.

To better capture the system's dynamics, we customized the model to use circular padding, which respects the periodic boundary conditions of the Gray-Scott simulation. The specific architecture, detailed in Listing 1, is a lightweight U-Net with a symmetric encoder-decoder structure. The encoder consists of three `DownBlock2D` blocks that progressively downsample the input, while the number of feature channels increases from 16 to 32, and finally to 64. The decoder mirrors this

structure with three `UpBlock2D` blocks to reconstruct the output. This streamlined design reduces computational overhead while effectively modeling the spatial features of the patterns.

```python
def get_unet_diffusion():
    unet = UNet2DModelWithPadding(
        sample_size=128,
        in_channels=2,
        out_channels=2,
        down_block_types=
            ("DownBlock2D", "DownBlock2D", "DownBlock2D"),
        up_block_types=
            ("UpBlock2D", "UpBlock2D", "UpBlock2D"),
        block_out_channels=(16, 32, 64),
        norm_num_groups=8,
        padding_mode='circular',
        only_when_effective=True,
        log_changed=True
    )
    return unet
```

Listing 1: Python code for configuring the 2D U-Net model. We customize the model to use circular padding, which respects the periodic boundary conditions of the Gray-Scott simulation.

### A.3 SCORE FUNCTION ESTIMATION

#### A.3.1 DIFFUSION MODELS

To estimate the score function $s(x) = \nabla \log p(x)$ for data sampled from $p(x)$, we utilized diffusion models (Ho et al., 2020; Song et al., 2020). These models work by training neural networks to recover the original data from noisy versions.

The training process involves minimizing the following loss function:

$$L = \mathbb{E}_{x \sim p(x), \tau \sim \mathcal{U}(0,1), \epsilon \sim \mathcal{N}(0, I_d)} \left\| \epsilon_\theta(\sqrt{\alpha_\tau} x + \sqrt{1 - \alpha_\tau} \epsilon, \tau) - \epsilon \right\|^2 \tag{17}$$

In this formulation, $x_\tau = \sqrt{\alpha_\tau} x + \sqrt{1 - \alpha_\tau} \epsilon$ represents a noisy version of the original data $x$, where $\alpha_\tau$ controls the noise level based on $\tau$. The noise parameter $\tau$ interpolates between the original distribution ($\tau = 0$, giving $x_0 \sim p(x)$) and pure Gaussian noise ($\tau = 1$, giving $x_1 \sim \mathcal{N}(0, I^d)$) (Nichol & Dhariwal, 2021).

Once trained, the model $\epsilon_\theta$ can predict the noise component added to the original data. From this noise prediction, we can derive the score function $s$ of $p(x_\tau)$ (Dhariwal & Nichol, 2021) using:

$$s(x_\tau) = -\frac{1}{\sqrt{1 - \alpha_\tau}} \epsilon_\theta(x_\tau, \tau). \tag{18}$$

For practical implementation, we set $\tau > 0$ rather than $\tau = 0$: at $\tau = 0$, the model would be untrained, while high $\tau$ values would deviate significantly from the original distribution. Our experiments show that $\alpha_\tau \approx 0.95$ provides good performance across most scenarios, which corresponds to $\tau = 0.2$ in 2D and Lorenz systems, and $\tau = 0.1$ for Gray-Scott patterns.

#### A.3.2 V-PREDICTION

While the original diffusion model training objective works well on toy models, it shows limitations on larger, more complex tasks. Specifically, the estimated score functions are not accurate enough for downstream applications. To address this, we adopt the $v$-prediction method (Salimans & Ho, 2022). Instead of training a model to predict $\epsilon$, which is the noise added to the data, we train it to predict the "velocity" defined as:

$$v = \sqrt{\alpha_\tau} \epsilon - \sqrt{1 - \alpha_\tau} x_0. \tag{19}$$

The model $v_\theta$ is trained to predict this velocity instead of the noise directly:

$$L = \mathbb{E}_{\boldsymbol{x} \sim p(\boldsymbol{x}), \tau \sim \mathcal{U}(0,1), \boldsymbol{\epsilon} \sim \mathcal{N}(0, \boldsymbol{I}_d)} \left\| v_\theta(\boldsymbol{x}_\tau, \tau) - \boldsymbol{v} \right\|^2. \tag{20}$$

This approach can significantly accelerate the training process and achieve higher accuracy in score estimation. Based on the design of diffusion models, we can derive:

$$\boldsymbol{\epsilon} = \frac{\boldsymbol{x}_\tau - \sqrt{\alpha_\tau} \boldsymbol{x}_0}{\sqrt{1 - \alpha_\tau}}. \tag{21}$$

Hence, we can replace $\boldsymbol{\epsilon}$ in equation 18 with the formula for $\boldsymbol{x}_0$:

$$s(\boldsymbol{x}_\tau) = -\frac{\boldsymbol{\epsilon}}{\sqrt{1 - \alpha_\tau}} \tag{22}$$

$$= \frac{\boldsymbol{x}_0 \sqrt{\alpha_\tau} - \boldsymbol{x}_\tau}{1 - \alpha_\tau}. \tag{23}$$

Therefore, we can unify the score function estimation via the estimated $\boldsymbol{x}_0$, which is independent of which prediction objective we use.

### A.3.3 BATCH NORMALIZATION

Since the distribution $p(\boldsymbol{x})$ is invariant under the flow $\boldsymbol{v}(\boldsymbol{x})$, the average position $\langle \boldsymbol{x} \rangle$ is constant. The time evolution of the average position is given by the average velocity, which must therefore be zero:

$$\frac{\mathrm{d}}{\mathrm{d}t} \langle \boldsymbol{x} \rangle = \langle \boldsymbol{v} \rangle = \boldsymbol{0}. \tag{24}$$

Based on this property, we enforce this zero-mean condition on our neural network $v_\theta$ by applying a batch normalization layer to its output. A standard batch normalization layer includes a learnable affine transformation that could produce the trivial solution $\boldsymbol{v} \equiv \boldsymbol{0}$ (e.g., by learning a zero scaling factor). To prevent this, we use a layer that only performs normalization. For a mini-batch of raw network outputs $\{\hat{\boldsymbol{v}}(\boldsymbol{x}_k)\}$, the final output $\boldsymbol{v}(\boldsymbol{x}_k)$ for each input $\boldsymbol{x}_k$ is computed as:

$$\boldsymbol{v}(\boldsymbol{x}_k) = \frac{\hat{\boldsymbol{v}}(\boldsymbol{x}_k) - \mathbb{E}_{\text{batch}}[\hat{\boldsymbol{v}}]}{\sqrt{\text{Var}_{\text{batch}}[\hat{\boldsymbol{v}}] + \epsilon}}, \tag{25}$$

where $\mathbb{E}_{\text{batch}}[\hat{\boldsymbol{v}}]$ and $\text{Var}_{\text{batch}}[\hat{\boldsymbol{v}}]$ are the mean and variance calculated over the current mini-batch. During inference, these are replaced by their corresponding running averages maintained throughout the training process.

## B LONG-TERM DISTRIBUTION PRESERVATION

To validate our method's ability to preserve distributions over extended time periods, we evaluate distribution preservation over time on the Lorenz system. We simulate the dynamics for 100 steps with step size $\Delta t = 0.1$ using the Runge-Kutta method and compute the Maximum Mean Discrepancy (MMD) (Gretton et al., 2006) between the initial distribution and distributions at different time steps.

As shown in Figure 7, our methods (both learned dynamics and the training-free approach) exhibit significantly lower MMD values, demonstrating that our methods can effectively preserve the distribution over long time horizons.

## C TURING PATTERNS

We use Gray-Scott model (Gray & Scott, 1984) to generate Turing patterns. The model is defined by the following reaction-diffusion equations:

$$\frac{\partial \boldsymbol{u}}{\partial t} = D_u \nabla^2 \boldsymbol{u} - \boldsymbol{u}\boldsymbol{v}^2 + F(1 - \boldsymbol{u}),$$
$$\frac{\partial \boldsymbol{v}}{\partial t} = D_v \nabla^2 \boldsymbol{v} + \boldsymbol{u}\boldsymbol{v}^2 - (F + k)\boldsymbol{v}. \tag{26}$$

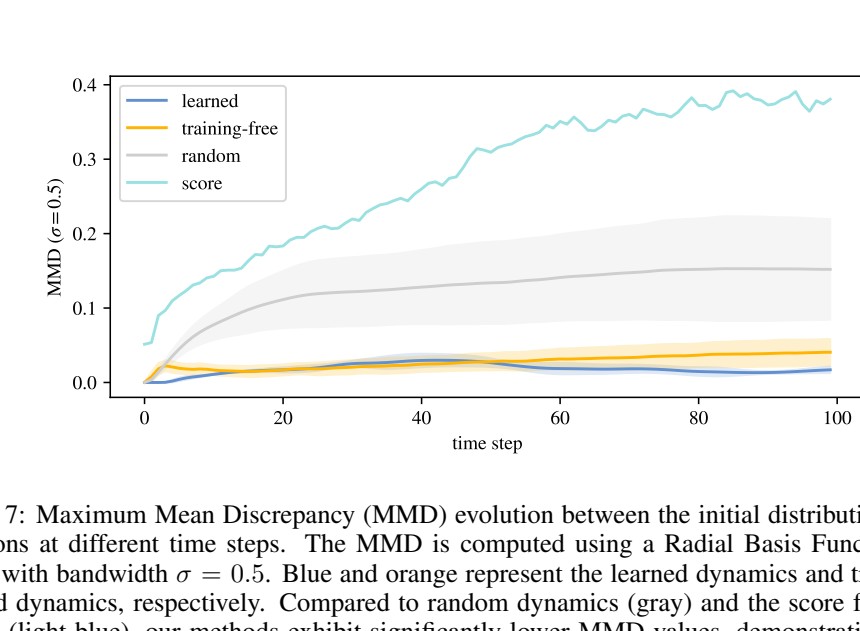

Figure 7: Maximum Mean Discrepancy (MMD) evolution between the initial distribution and distributions at different time steps. The MMD is computed using a Radial Basis Function (RBF) kernel with bandwidth $\sigma = 0.5$. Blue and orange represent the learned dynamics and training-free method dynamics, respectively. Compared to random dynamics (gray) and the score function approach (light blue), our methods exhibit significantly lower MMD values, demonstrating superior long-term distribution preservation.

By changing the hyperparameters $D_u$, $D_v$, $F$, and $k$, we can generate a variety of Turing patterns. Here, we select three sets of parameters that produce distinct patterns, with $D_u = 0.16$, $D_v = 0.08$, and other parameters varying:

- "life": Life-like pattern: $F = 0.006$, $k = 0.045$.
- "wave": Wave pattern: $F = 0.018$, $k = 0.049$.
- "spirals": Spiral pattern: $F = 0.007$, $k = 0.028$.
- "maze": Static pattern: $F = 0.029$, $k = 0.057$.

## D TRAINING-FREE METHOD

### D.1 SKEW-SYMMETRIC SOLUTION

**Theorem 1.** *The vector field* $\boldsymbol{v}_S(\boldsymbol{x}) = \boldsymbol{S}\nabla \log p(\boldsymbol{x})$, *constructed with any constant skew-symmetric matrix* $\boldsymbol{S}$ ($\boldsymbol{S}^\top = -\boldsymbol{S}$), *is a solution to the steady-state continuity equation,* $\nabla \cdot [p(\boldsymbol{x})\boldsymbol{v}(\boldsymbol{x})] = 0$.

*Proof.* $\nabla \cdot [p(\boldsymbol{x})\boldsymbol{v}(\boldsymbol{x})] = 0$ can be simplified by noting that $\nabla p(\boldsymbol{x}) = p(\boldsymbol{x})\nabla \log p(\boldsymbol{x})$, which leads to:

$$\nabla \cdot [p(\boldsymbol{x})\boldsymbol{S}\nabla \log p(\boldsymbol{x})] = \nabla \cdot \left[p(\boldsymbol{x})\boldsymbol{S}\frac{\nabla p(\boldsymbol{x})}{p(\boldsymbol{x})}\right] = \nabla \cdot [\boldsymbol{S}\nabla p(\boldsymbol{x})].$$

Using Einstein notation, this divergence is a contraction between $\boldsymbol{S}$ and the Hessian of $p(\boldsymbol{x})$. This term vanishes due to the interplay between the skew-symmetry of $\boldsymbol{S}$ (i.e., $S_{ij} = -S_{ji}$) and the symmetry of the second partial derivatives of $p(\boldsymbol{x})$ (i.e., $\partial_i\partial_j p = \partial_j\partial_i p$). The derivation is as follows:

$$\nabla \cdot [\boldsymbol{S}\nabla p(\boldsymbol{x})] = S_{ij}\partial_i\partial_j p(\boldsymbol{x})$$
$$= \frac{1}{2}\left(S_{ij}\partial_i\partial_j p(\boldsymbol{x}) + S_{ji}\partial_j\partial_i p(\boldsymbol{x})\right)$$
$$= \frac{1}{2}\left(S_{ij}\partial_i\partial_j p(\boldsymbol{x}) - S_{ij}\partial_i\partial_j p(\boldsymbol{x})\right) = 0.$$

In the second line, we rewrite the term by averaging it with a copy where the indices $(i, j)$ have been swapped. In the third line, we substitute the aforementioned properties. $\square$

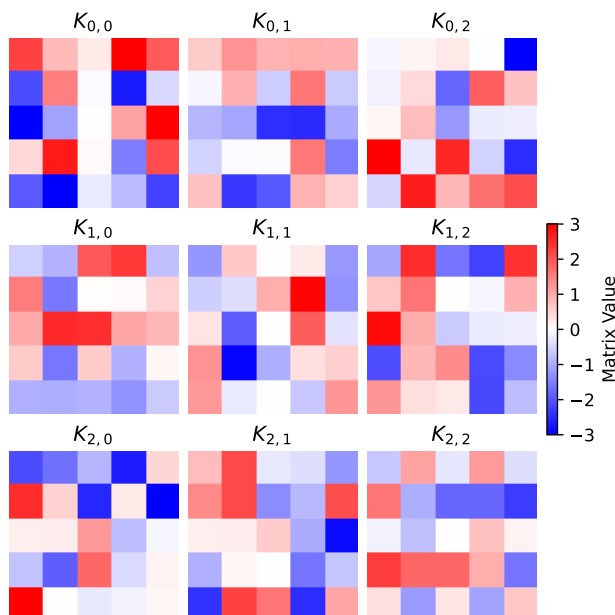

Figure 8: An example of a kernels with $c = 3$ channel, $5 \times 5$ convolutional layer satisfying the skew-symmetry condition. Note how the off-diagonal kernel $K_{12}$ is the negated and spatially flipped version of $K_{21}$, and the diagonal kernel $K_{22}$ is point-symmetric with negation (e.g., the top-left value is the negative of the bottom-right value).

## D.2 SKEW-SYMMETRIC CONVOLUTIONAL LAYERS

**Theorem 2.** *A convolutional layer with kernel* $K \in \mathbb{R}^{c \times c \times (2r+1) \times (2r+1)}$ *corresponds to a skew-symmetric matrix if and only if its weight tensors satisfy the condition*

$$K_{o,i,u,v} = -K_{i,o,-u,-v} \quad \forall o, i \in \{1, \dots, c\}, \quad \forall (u,v) \in \{-r, \dots, r\}^2. \tag{27}$$

*Proof.* A linear transformation on a $c$-channel image of size $h \times w$ can be represented by a large block matrix $S \in \mathbb{R}^{(chw) \times (chw)}$ with $c \times c$ blocks. When this transformation is a standard convolution with a kernel $K \in \mathbb{R}^{c \times c \times (2r+1) \times (2r+1)}$, the matrix $S$ is a block matrix given by:

$$S = \begin{bmatrix} C(K_{1,1}) & C(K_{1,2}) & \cdots & C(K_{1,c}) \\ C(K_{2,1}) & C(K_{2,2}) & \cdots & C(K_{2,c}) \\ \vdots & \vdots & \ddots & \vdots \\ C(K_{c,1}) & C(K_{c,2}) & \cdots & C(K_{c,c}) \end{bmatrix}, \tag{28}$$

where $K_{o,i} \in \mathbb{R}^{(2r+1) \times (2r+1)}$ is the kernel slice that maps the $i$-th input channel to the $o$-th output channel, and $C(K_{o,i}) \in \mathbb{R}^{(hw) \times (hw)}$ is the corresponding matrix for this single-channel convolution.

For the transformation $S$ to be skew-symmetric, it must satisfy the condition $S = -S^\top$. Since the transpose of a block matrix is given by $(S^\top)_{o,i} = S_{i,o}^\top$, the skew-symmetry condition requires that each block satisfies:

$$C(K_{o,i}) = -C(K_{i,o})^\top \quad \forall o, i \in \{1, \dots, c\}. \tag{29}$$

To translate this matrix equation into a direct constraint on the kernel weights, we examine the structure of $C(K)$. Under standard convolution definitions, the element of $C(K_{o,i})$ corresponding to the influence of an input at spatial location $q$ on an output at location $p$ is given by the kernel value at the displacement $p - q$. That is, $[C(K_{o,i})]_{\phi(p),\phi(q)} = (K_{o,i})_{p-q}$, where $\phi$ is an arbitrary bijection that maps the spatial location to the index of the matrix.

Consequently, the corresponding element of the transposed matrix is:

$$[C(K_{i,o})^\top]_{\phi(p),\phi(q)} = [C(K_{i,o})]_{\phi(q),\phi(p)} = (K_{i,o})_{q-p}. \tag{30}$$

Substituting these expressions into Equation 29 yields:

$$(K_{o,i})_{\boldsymbol{p}-\boldsymbol{q}} = -(K_{i,o})_{\boldsymbol{q}-\boldsymbol{p}}.$$

By defining the kernel coordinate vector as $\boldsymbol{u} = \boldsymbol{p} - \boldsymbol{q}$, we have $-\boldsymbol{u} = \boldsymbol{q} - \boldsymbol{p}$. The condition on the kernel weights thus simplifies to:

$$(K_{o,i})_{\boldsymbol{u}} = -(K_{i,o})_{-\boldsymbol{u}} \tag{31}$$

for all kernel coordinates $\boldsymbol{u} \in \{-r, \dots, r\}^2$. In index notation, with $\boldsymbol{u} = (u, v)$, this is written as:

$$K_{o,i,u,v} = -K_{i,o,-u,-v}. \tag{32}$$

This simple relation enforces skew-symmetry. For diagonal blocks ($i = o$), the kernel $K_{i,i}$ must be point-symmetric with negation ($K_{i,i,u,v} = -K_{i,i,-u,-v}$). For off-diagonal blocks ($i \neq o$), the kernel $K_{o,i}$ must be the negated and spatially flipped version of the kernel $K_{i,o}$. Figure 8 illustrates an example of a kernel that satisfies this property. $\qquad\square$

## E  ARTIFICIAL LIFE

### E.1  GENERATING TRAINING DATA

We generate training data by randomly scattering target patterns in a $128 \times 128$ 2D space with random rotations and positions. The space is circular to avoid boundary effects. To prevent pattern overlap, we use PNG format with alpha channels to represent pattern transparency, employing a trial-and-reject approach for pattern placement. We accept a pattern only if its alpha channel does not overlap with existing patterns. For each $128 \times 128$ image, we attempt to add patterns multiple times, with 2 placement attempts per pattern.

## F  THE USE OF LLM

We use LLMs to help us proofread the manuscript, including grammar correction and sentence structure improvement. We also use LLMs to assist with writing code that can be easily verified. For example, in the Artificial Life experiments, we use LLMs to write the code for generating training data, which we can easily verify for correctness by inspecting both the results and the code. LLMs also help us with well-defined mathematical problems. However, all proofs and mathematical derivations are verified by the authors.

