# OpenReview forum: "Equilibrium flow: From Snapshots to Dynamics"
_ICLR.cc/2026/Conference — Submitted to ICLR 2026_

### Official Review · Reviewer_9gCf · 2025-10-27

**Soundness:** 3
**Presentation:** 3
**Contribution:** 2
**Rating:** 8
**Confidence:** 2

**Summary:**

This paper introduces the _Equilibrium flow_ method for inference of the underlying continuous dynamics of a system from a distribution of static snapshots, without any temporal information. The principal idea is to recover a vector field that preserves the distribution of a given static pattern. The authors demonstrate the method's effectiveness on 2D systems, the chaotic Lorenz attractor, and high-dimensional Turing patterns from the Gray-Scott model, for which they also develop an efficient training-free variant. Beyond recovering known dynamics, the framework enables a new "inverse design" paradigm for creating Artificial Life by specifying a target pattern and discovering the local interaction rules that preserve it and lead to emergent behaviors like flocking.

**Strengths:**

- **Originality**:  addresses an ongoing challenge / fundamental problem --  inferring dynamics from static data -- in a novel way; avoiding the compute intensive task of finding dynamics for a single static pattern, and focusing on distributions.  In addition, the application to inverse design (Artificial Life) is creative and demonstrates the potential of the framework beyond tyoical system recovery.

- **Quality**: the work is of good quality -- coupling theoretical foundations and motivation with empirical evaluations. The mathematical derivations are clearly explained and easy to follow, and the experiments span a wide diversity of systems. Specifically the effort on the methods' relevance, applicability and value is highly appreciated.

- **Clarity**: The paper is very well-written and easy to follow. The motivation is clearly established, and the methodology is presented logically. The appendix, along with supplementary code assist following the framework and grasping the papers' contribution. At last, the discussion nicely summarized the paper and outlines future direction.

- **Significance**: The method is of high value as it addresses a setting which is prevalent in many scientific domains--scarce time-series data yet abundant static snapshots. In addition, as indicated by the authors, the inverse-design approach could be used in future work concerning artificial life formation.

**Weaknesses:**

- **Quantitative evaluation**: while the authors presented a diverse set of experiments it will be valuable to provide additional quantitative evaluation, supporting the qualitative results, especially for the training-free setting.
- **Hyper-parameters selection**: while the existence of hyper-parameters is inevitable, for practical reasons it will be valuable to present how these shall be set / tuned, the robustness to these or alternatively their impact.
- **Limitations and failure modes**: the paper is lacking a discussion on the limitations of the approach and potential failure modes.

**Questions:**

Following the above weaknesses:
1. would the authors be able to provide additional quantitative assessment of performance, including failure points and limitations.
2. relate to hyper-parameters defined and used.
3. discuss existing limitations.

---

> ### Author Response · Authors · 2025-11-17
>
> Thank you for the constructive suggestions and questions.
>
> 1. **Hyperparameters:**
> For the original Hutchinson-based method, our work involves (1) training a diffusion model to estimate the score, and (2) training the flow model. For the diffusion model, we follow standard hyperparameter choices commonly used in diffusion models to ensure high score accuracy—this is a well-studied setting, so we adopt established conventions. For the flow model, the main hyperparameters are the number of trace-estimation samples $k$ and the diffusion time $\tau$. Empirically, we find that using a small but non-zero $\tau$ (equivalent to $\alpha_\tau = 0.95$) provides a good balance between efficiency and fidelity. For the trace-estimation samples, we observe that $k = 4$ is typically sufficient. We will add further discussion in the revised version.
>
> For the training-free method, the key hyperparameter is the skew-symmetric matrix $S$, where $S = -S^\top$, used to define the flow $v(x) = S \nabla \log p(x)$. For vector-based data without temporal information, we recommend the Hutchinson-based method. For image data, although both $S$ and the corresponding convolutional kernel $K$ admit many choices, we suggest using a $1 \times 1$ kernel, which preserves the inductive bias of the trained diffusion model. Larger kernels would introduce additional asymmetry that we wish to avoid.
>
> 2. **Limitations:**
> Thank you for highlighting this. We will incorporate a dedicated discussion in the revised version. In short, our method relies on two key assumptions: (1) the **equilibrium assumption**, meaning the system is sampled in an equilibrium state; and (2) the **uniform sampling assumption**, meaning the samples are drawn uniformly without systematic bias. If either assumption is violated, the method may not perform well. Fortunately, both assumptions are often verifiable in practice.
>
> 3. **Quantitative analysis:**
> We are actively adding more quantitative experiments, including cross-method comparisons between the training-based and training-free approaches. We will also include quantitative tests evaluating the equilibrium assumption using empirical data.

---

### Official Review · Reviewer_hmyh · 2025-10-30

**Soundness:** 2
**Presentation:** 3
**Contribution:** 2
**Rating:** 2
**Confidence:** 4

**Summary:**

This paper proposes Equilibrium flow, a method to infer continuous dynamics that preserve a given static pattern distribution—without requiring any temporal information. The core idea leverages the steady-state continuity equation $\nabla \cdot \[p(x)v(x)\] = 0$, which is reformulated into a local constraint $\nabla \cdot v(x) + v(x)\cdot s(x) = 0$ using the score function $s(x) = \nabla \log p(x)$. The authors demonstrate applications on 2D distributions, the Lorenz chaotic attractor, high-dimensional Gray-Scott Turing patterns (via a training-free variant), and even "inverse design" of artificial life forms. A key claim is that the pattern distribution strongly constrains the space of possible dynamics, and that the learned dynamics exhibit high fidelity to ground-truth systems.

**Strengths:**

- The basic problem (from snapshots to dynamics) is interesting, with potential applications in several scientific fields.
- The training-free variant for high-dimensional data—based on skew-symmetric transformations of the score—is a practical and computationally efficient adaptation of known irreversible Langevin dynamics.
- Broad experimental scope and strong visual results.
- Overall I found the paper well-written in terms of organization, flow, among other aspects.

**Weaknesses:**

1. Ill-posed problem misrepresented: The inverse problem "recover $v$ from $p$ " is severely underdetermined, yet ill-posed. The condition $\nabla \cdot(p v)=0$ is a single scalar PDE for a $d$-dimensional vector field $v$. In $\mathbb{R}^d$, the solution space is infinite-dimensional. By Helmholtz decomposition, $v=-\nabla \phi+\nabla \times A$,the continuity equation only constrains the irrotational part via $\phi=\log p+\operatorname{const}$; the solenoidal part $\nabla \times A$ is completely free. Thus, any divergence-free flow on level sets of $p$ is admissible. The paper acknowledges this in passing (Sec. 4) but frames the problem as if the data "strongly constrains" dynamics-this is misleading without additional physical priors.

2.  Singular Invariant Measures:
For the Lorenz system, the invariant measure $\mu$ is singular with respect to Lebesgue measure: it is supported on a fractal attractor of Hausdorff dimension $\approx 2.06$. Consequently, $p(x)=d \mu / d x$ **does not exist**, and $\nabla \log p(x)$ is mathematically undefined.
The diffusion model instead estimates the score of a smoothed measure $p_\tau=\mu * \mathcal{N}\left(0, \tau^2 I\right)$, which has a smooth density. But there is no theoretical guarantee that the dynamics derived from $s_\tau(x)=\nabla \log p_\tau(x)$ approximate the true Lorenz vector field $v_{\text {Lorenz }}(x)$. In fact, as $\tau \rightarrow 0, s_\tau(x)$ becomes increasingly noisy and uninformative off the attractor, making the learned $v(x)$ unreliable.

3. Fidelity metrics are misleading: Cosine similarity of 0.25 ± 0.01 (Table 1) for Lorenz is labeled “high fidelity,” but corresponds to ~75° average angle—far from alignment.

Moreover, chaotic systems are not characterized solely by positive  Lyapunov exponents. The Lorenz system has a specific Lyapunov spectrum $\left(\lambda_1 \approx 0.91, \lambda_2=0, \lambda_3 \approx-14.57\right)$, **a strange attractor topology**, and a **kneading invariant**. None of these are verified. The learned system could be any chaotic flow (e.g., Rössler), and the paper provides no evidence it is topologically conjugate to Lorenz.


4. Gray–Scott validation is insufficient: The training-free method uses a $1 \times 1$ convolutional kernel (Eq. 13), i.e., a **pointwise linear transformation**
$$
v_S(x)=\gamma\left[\begin{array}{cc}
0 & -1 \\
1 & 0
\end{array}\right] s(x)
$$
This cannot represent the Laplacian terms $\nabla^2 u, \nabla^2 v$, which are non-local (require spatial neighborhoods). Thus, the learned $v(x)$ cannot satisfy the Gray-Scott PDE structure. The visual similarity in Figure 3 is insufficient; *the method recovers some distribution-preserving flow, but not the reaction-diffusion dynamics.*

5. Uniqueness misattributed: Figure 4A shows neural solutions cluster tightly while training-free ones scatter—yet the paper claims “data strongly constrains dynamics.” In fact, clustering stems from neural inductive bias (smoothness, simplicity), not data (as even the authors note on p.7).

**Questions:**

1. On Lorenz fidelity: Given that the cosine similarity is only 0.25 , what dynamical invariants (e.g., Lyapunov exponents, fractal dimension, Poincaré return maps) confirm that the learned system is specifically the Lorenz system rather than an arbitrary chaotic flow?
2. On singular measures: Since the Lorenz invariant measure has no density, how do you justify using $\nabla \log p(x)$ ? What is the relationship between the smoothed score $s_\tau(x)$ and the true vector field $v_{\text {Lorenz }}(x)$ in the limit $\tau \rightarrow 0$ ?
3. On uniqueness: Why do training-free solutions exhibit higher variance than neural network solutions if the data strongly constrains dynamics? Doesn't this imply that "uniqueness" is due to model bias, not data?
4. On Gray-Scott: How can a $1 \times 1$ convolution capture the non-local Laplacian terms $\nabla^2 u$ ?
5. On well-posedness: What minimal additional assumptions (e.g., smoothness, locality, conservation laws) would make the inverse problem well-posed? Have you considered incorporating such priors?
6. On the use of MMD for long-term validation: Figure 7 shows low MMD over time, but MMD is insensitive to dynamical structure—it only measures marginal distribution similarity. Could two systems with identical invariant measures but different ergodic properties (e.g., mixing rates, recurrence times) yield the same MMD?
7. On scalability: How does the method scale to higher-dimensional systems or systems with more complex dynamics？

---

> ### Author Response · Authors · 2025-11-16
> **Reply on the weaknesses**
>
> Thank you for your detailed reading and insightful feedback. We will address each point one by one.
>
> 1.  **Regarding the ill-posed problem:** Yes, the problem is ill-posed, which is exactly why this problem is both important and challenging. However, this does not mean we are unable to find potential solutions. As discussed in our section on uniqueness, the **smoothness bias** of the neural network constrains the space of possible solutions. As a framework, our method does not preclude adding **additional constraints** (e.g., adding new loss terms or using a specifically designed network architecture).
>
> 2.  **Regarding the singular invariant measure:** This is one of the reasons we chose a non-zero $\tau$ for our experiments. We **empirically selected** a small but non-zero $\tau$ to balance smoothness and fidelity. Our experiments on the Lorenz system are not intended to show that our method can exactly recover the ground truth, as we discussed in the paper and as you correctly noted. Rather, we want to use this dataset to demonstrate that the data distribution *does* constrain many features of the system, including its chaotic features. We will add experiments with more qualitative measurements for the Lorenz system in the revision.
>
> 3.  **Regarding the $1 \times 1$ kernel:** This is a very keen observation. This was an **intentional design choice**, and it does not result in a loss of capability for capturing the Laplacian term. As we defined, $v=S \nabla \log p(x)=S s(x)$. Although a $1 \times 1$ kernel does not capture local structures *itself*, the trained score function (i.e., the diffusion model) has a much larger receptive field due to its convolutional layers in the UNet architecture. This leads to *why* we chose a $1 \times 1$ kernel over larger ones: a $1 \times 1$ kernel does not add further asymmetry, thus **inheriting the inductive bias** directly from the pre-trained diffusion model. As we discussed regarding uniqueness, the final result is dually constrained by the data distribution and this neural network bias.
>
> 4.  **Regarding Uniqueness:** Thank you for the detailed reading. We agree that inductive bias has a very important effect. However, we consider this to be only one part of the story, as the data distribution also plays a critical role. For example, when the *same* MLP architecture is trained on the two-moons data versus the two-peak data, it learns very different behaviors. This alone demonstrates how the data distribution constrains the potential dynamics. Therefore, we concluded in the paper that this constraint arises from *both* the data distribution and the inductive bias.

---

> > ### Comment · Reviewer_hmyh · 2025-11-27
> >
> > Thank you for the updated manuscript and detailed rebuttal. I’ve read it carefully. While the clarifications on snapshot-only motivation, ill-posedness/inductive bias, the use of a small non-zero (\tau) for Lorenz, and the design of the skew block improve scope and readability, my core concerns remain unresolved, so I will maintain my original score.
> >
> > To sharpen the paper, I suggest three concise additions in the future version:
> > 1. a short, explicit statement of **applicability boundaries** (e.g., clearly non-equilibrium cases where the equilibrium assumption is inappropriate);
> > 2. clearer **positioning vs. methods using sparse time or pseudo-time**, so readers understand when those alternatives are preferable;
> > 3. **minimal dynamics-aware validation**, e.g., invariant-level checks for Lorenz beyond a single LE and basic PDE-structure tests for Gray–Scott.
> >
> > These edits would make the contributions and limitations more precise and align with ICLR’s emphasis on clear context and concrete evidence.

---

### Official Review · Reviewer_tfX3 · 2025-10-31

**Soundness:** 2
**Presentation:** 4
**Contribution:** 2
**Rating:** 2
**Confidence:** 4

**Summary:**

The authors present a method for learning “equilibrium flows” under which a given distribution of data is invariant. They are motivated by the problem of snapshot modeling, in which a dynamical system leaves a static trace of an underlying dynamical process. They take a standard continuity-equation approach with the technical innovation of approximating the divergence term through sampling. This is paired with a training-free method which uses an extension of Langevin Monte Carlo dynamics. The equilibrium flows approach is then applied to a collection of synthetic data sets, starting with low-dimensional but still interesting point clouds, turning next to reaction-diffusion PDEs and ultimately to images of repeated tokens which fall for the authors under the rubric of “artificial life”. They evaluate their results both qualitatively–for example, by how well their learned flows preserve the coarse structure of the point clouds or whether a flow learned from a chaotic system remains chaotic–and quantitatively via similarity scores between flows learned in various competing ways. The authors attribute the performance of their neural-network-based method to certain basic inductive biases encouraged by such an architecture.

**Strengths:**

1. Snapshots are an interesting data type, as the authors convincingly argue, and the field needs more methods for modeling the underlying generative dynamic model. Also, I think that their focus on preservation of qualitative features of the underlying dynamics is interesting and goes beyond more common approaches for merely predicting specific time series or end-states.

2. The qualitative results on 2-3d point clouds and reaction diffusion dynamics mostly look quite good, and it’s nice to see, for example, that the Lorenz case doesn’t blow up and deviations between nearby trajectories seem similar to the underlying model.

3. I thought the sampling approach via the Hutchinson estimator was interesting and differentiated their approach from other flow learning methods, though, as I will explain later, I don’t think this was explored deeply enough.

**Weaknesses:**

*Areas for improvement*

1. The main technical novelty of the paper, to my eye, seems to be the efficient way of computing the divergence. Otherwise, replacing Eq. 8 with a non-sampling approach would more or less be a reproduction of established methods. However, I don’t think the sampling approach is justified well. It is introduced as a way of estimating the divergence, which “is computationally expensive for high-dimensional x”. The authors claim they can “efficiently approximate it using Hutchinson’s Trace Estimator.” Yet, they seem to only use this estimator in the low-dimensional cases where div(v) would seem to be computable without sampling. Further, when they turn to those aforementioned high-dimensional cases, they do away with the Hutchinson estimator, saying  “...applying our original method to such high-dimensional data is challenging due to the computational cost of Hutchinson’s trace estimator.” This seems like a contradiction with the earlier quote. The authors need to clarify (1) when the estimator is useful, ideally by showing numerical results on performance as dimensionality grows and (2) if the use of estimator is not the sole novelty of the approach, where that novelty lies.


2. I think the quantitative analysis and evaluation could be greatly strengthened and applied more consistently. Most importantly, I think the authors should show quantitatively that the flows they learn are indeed equilibrium flows. This should be straightforward: perhaps they could simply push their data forward in time and show that some MMD does not increase much (as they do in the appendix)? It is not clear to me that this is even true. For instance, flow for the “moons” data in Fig 2a seems like it would destroy that distribution but this is hard to tell by eye. I would like to see this sort of distributional stability analysis applied consistently through the paper–assuming this is indeed what the authors want to show. Furthermore, the comparison between learned and training-free is not carried out consistently. If a flow could be learned in a training-free way for the 2d systems in Fig 2a, shouldn’t we compare those two approaches? This could be facilitated by the sort of quantitative metrics I proposed.


3. From Sec. A.3.3., it seems like the variance rescaling is important for preventing the model from converging to the zero flow. Could the authors verify this explicitly with a simple ablation experiment–batch normalization on vs off during training?


4. I am not yet convinced by artificial life example. The images look cool, but it’s not clear how to evaluate the results: are those squiggles on the image tokens in Fig 5 good or bad? Maybe you could generate these with ground truth dynamics and then try to recover it?


5. The training-free method seems like a restatement of earlier work on Langevin MC. If not, the authors should explain the novelty. Again, I think it should be compared systematically to the training-based approach, ideally on a performance metric that relates to equilibrium or the preservation of the distribution.


6. If I’m reading Sec. 3.1 correctly, the authors only use a simple rotation matrix for a convolutional kernel. But Fig. 8 shows many different kernels. Could the authors show the effect of different K? Could K be learned?


7. I think that the paper would be greatly strengthened with the inclusion of real data. For example, snapshot data is prevalent in single-cell transcriptomic studies. Could the authors show that they learn the cell cycle using data from, e.g., Riba et al., 2022 (Nat. Commun.)? This is just a suggestion, and really any real data would suffice–I just think that single cell data is a great use case for this approach.

*Small remarks*

Fig 4 caption says the training-free solutions are in green, but for me they show up as yellow.

**Questions:**

To restate some questions I hinted at above:

1. Can the authors clarify (1) when the Hutchinson estimator is useful, ideally by showing numerical results on performance as dimensionality grows and (2) if the use of estimator is not the sole novelty of the approach, where that novelty lies. In short, can you try to resolve the contradiction I pointed out above?

2. Could you include MMD scores or something else that demonstrates the distributions are indeed preserved? This should be ideally done for all data sets.

3. Could you consistently compare (again via a metric like in point 2 above) the training-free vs learned approach? For now, the similarity scores are not so convincing. If you are learning equilibrium flows, you have to show either that the flows are the "true" flows given an underlying true dynamics or that the flows do preserve the distribution or both.

4. Can you show results with and without batch normalization?

5. Is it possible to show that the method recovers the true generative dynamics of your artificial life examples? If this is not possible, I think you will have to show in some other way why this example is technically interesting.

6. What is the novelty of the training-free approach?

7. What is the effect of different K? Could K be learned?

8. Have the authors thought of trying real data? Could this be used for the single-cell data I mentioned above? On its own, a detailed study on real data (even with the bulk of results in the appendix if necessary) would be enough for me to reconsider my scores.

---

> ### Author Response · Authors · 2025-11-16
> **First round reply**
>
> We thank the reviewer for their insightful suggestions. Given the time required to run the new experiments, we will first address the questions that can be answered without them. We are actively working on the suggested experiments and will include the results in our updated paper.
>
> 1.  **Hutchinson estimator:** Thank you for highlighting this point. We will clarify the motivation for our training-free method in the revision. We *did* apply the Hutchinson estimator to high-dimensional systems like Turing patterns (although not included in the paper). With some random seeds, this approach produced dynamics close to the ground truth, but the training was very time-consuming (approx. 8 hours per experiment on a single NVIDIA A6000 GPU). We will attempt to extend the training time to assess if this leads to more stable results.
>
>     This high computational cost directly motivated our exploration of the **training-free method**. Therefore, the two methods are not contradictory; rather, the training-free method is presented as a practical **trade-off**. It is significantly faster (only requiring a trained diffusion model) but has a less natural inductive bias. We will make this motivation and the trade-off between the two methods explicit in the revised paper.
>
> 2.  **the "moons" data:** Yes, we performed the suggested experiment for the Lorenz system (presented in the appendix), which shows that the distribution is preserved. The reviewer makes a keen observation regarding the two-moons data. We acknowledge that the learned flow does not perfectly preserve the two-moon distribution, which we attribute to the smoothing effect of our inductive bias.
>
>     However, our loss function (Eq. 8) not only minimizes the global divergence $\nabla\cdot [p(x)v(x)]$ but also estimates *local* divergence. As shown in the colorized plot (see anonymous link), while most data points have near-zero divergence, the **edges of the distribution exhibit non-zero divergence**. We consider this a valuable feature of our method: it not only finds the potential dynamics but can also detect potential **sources and sinks** in the data. We will highlight this capability as a "side product" and discuss its potential in the updated version.
>     [source and sinks](https://i.postimg.cc/hGYTFVd3/output.png)
>
> 3.  **ablation on variance rescaling (Batch Normalization):** We did perform an experiment to verify this. As an alternative to using batch normalization to prevent the zero-flow solution, we tested a modified loss $\mathcal L' = \mathcal L / |v_\theta(x)|^2$ and removed the BN layer. This modified loss also successfully prevented convergence to the trivial zero solution (though it was harder to train).
>
>     Using this setup, we monitored the original loss $\mathcal L$ versus the average $v_\theta(x)$, as shown in the new figure. We observed a strong correlation: as the original loss $\mathcal L$ decreased, $v_\theta(x)$ also converged to zero. This ablation confirms that zero-v assumption is valid.
>     [compare.png](https://i.postimg.cc/XYQDC1T1/compare.png)
>
> 4.  **artificial life:** We will clarify the goal of this experiment. The "Artificial Life" domain is often focused less on modeling *existing* life and more on exploring "what life *could* be." A key challenge is discovering dynamics that generate specific, complex patterns while also allowing for diversity. Existing methods typically rely on evolutionary search algorithms to find these dynamics.
>
>     The novelty we demonstrate is a **generative method** that can directly produce dynamics matching a desired target pattern (the "squiggles" in Fig 5) **without requiring any search process**. We will make this distinction and contribution clearer in the revision.
>
> 5.  **Regarding Langevin MC:** Yes, our training-free method is inspired by work on Langevin MC, which we cited in the paper. We agree that a direct comparison is needed. We will add quantitative comparisons between our training-free and training-based approaches, evaluated on metrics related to equilibrium, in the updated version.
>
> 6.  **Regarding the kernels in Fig. 8:** Thank you for the detailed question. Figure 8 is intended to illustrate the *potential* symmetry structures of different kernels. We used a simple $1 \times 1$ rotation matrix in our experiments for a specific and important reason: **to preserve the rotational symmetry** learned by the diffusion model, which we know is inherent to the Turing pattern dynamics. Using a more complex, learnable kernel $K$ would break this learned symmetry and impose a different, incorrect inductive bias. We will clarify this deliberate design choice in the text.
>
> 7.  **Regarding real data:** Thank you for this excellent suggestion. We agree that applying our method to real-world data, such as the single-cell transcriptomics data you suggested, would greatly strengthen the paper. We are actively working on this and hope to include preliminary results in the final version.

---

> > ### Comment · Reviewer_tfX3 · 2025-11-23
> >
> > Thank you very much for the point-by-point reply to my remarks. I am looking forward to seeing the revised manuscript, if possible.
> >
> > Your reply goes some way in addressing my critique, but there are a few points of continued concern on my part which I'd like to see addressed in the final version and in your comments here:
> >
> > 1. **Hutchinson**: Yes, it makes sense to maybe have a faster training-free version and a slower training-based version, but I still think that the apparent contradiction in the MS needs to be addressed. You say that Hutchinson is used to alleviate numerical difficulties, yet your reply here and the current wording in the MS makes it seem like it in fact introduces numerical problems. For example, if the training-free (i.e. Langevin MC) approach works well on Gray-Scott, then why wouldn't it work for the simpler data sets? If that is the case, you'd need to justify why your training-based (i.e. sampling) approach is useful.
> >
> > 2. **Moons**: It makes sense to me that you would find sources and sinks which help equilibrate your data, and that's cool. And it's also reasonable that your solutions would deviate from equilbrium in some cases, but you should maybe try to explain why. For instance, it could relate to the expressiveness of however you parameterize v or numerical issues. I don't see why moon-shaped limit cycles or other periodic orbits couldn't stabilize your moons except for the fact that those orbits would be twisty and hard to learn for engineering reasons.
> >
> > 3. **Ablation** I like the new loss, but, yes, it looks unstable on its face. If in practice you don't converge to the zero solution because of BN, then I think it suffices to emphasize that in the text without necessarily introducing a new loss. I would just want to see that you get the zero flow when BN is removed or to understand why the flow is still non-trivial even without it.
> >
> > 4. **Artificial life** I understand the motivation, but I just find the term "artificial life" to be too grand for the level of the results currently presented. To me, it just looks like you are doing the same thing but on images which look superficially like moving creatures. My minimum threshold for calling this artificial life would be to have at least some true dynamical model underlying the agents which you should you can reconstruct. Otherwise, there is no way to evaluate the quality of these results.

---

### Official Review · Reviewer_UXYh · 2025-11-02

**Soundness:** 3
**Presentation:** 3
**Contribution:** 3
**Rating:** 8
**Confidence:** 1

**Summary:**

This paper introduces Equilibrium Flow, a framework for inferring continuous dynamics directly from static pattern distributions, without temporal data. Using score-based neural representations, it reconstructs plausible dynamics across systems ranging from 2D flows to chaotic attractors and high-dimensional Turing patterns. The method further supports inverse design, enabling the discovery of local interaction rules that generate user-specified, life-like patterns.

**Strengths:**

1. The proposed method is grounded in solid theoretical foundations. Its connection to the continuity equation, score-based modeling, and Helmholtz decomposition is well-motivated and mathematically coherent.
2. The method is thoroughly evaluated on diverse systems: 2D distributions, the chaotic Lorenz system, and Gray-Scott Turing patterns, effectively demonstrating the method’s versatility.

**Weaknesses:**

The paper only presents successful cases, without discussing when or why the method fails (e.g., under noisy data, non-stationary distributions, or ambiguous patterns). This limits understanding of the robustness and failure boundaries of the approach.

**Questions:**

See weaknesses.

---

> ### Author Response · Authors · 2025-11-17
>
> Thank you for the insightful observation. We will incorporate this point into the revised version. In short, we acknowledge both the limitations and the scope of our method. Our approach relies on a key hypothesis: the observed data are uniformly sampled from an equilibrium state. Therefore, if the uniform-sampling or equilibrium assumptions are violated, the learned dynamics may deviate significantly from the ground truth. We will discuss this in more detail in the updated version.

---

### Official Review · Reviewer_8dfP · 2025-11-07

**Soundness:** 2
**Presentation:** 2
**Contribution:** 2
**Rating:** 4
**Confidence:** 3

**Summary:**

The idea of learning how data points move within a distribution without changing its overall shape by inferring a continuous vector from static samples using a pretrained diffusion model and neural network is reasonable. However, the reviewer has some concern regarding the assumption, motivation and performance of the proposed method, the detailed comments are as follows:
1) the authors state that the related methods are either computationally expensive or requiring the temporal information, however, the results does not really show how the proposed method avoid these.
2) the authors did not clarify why not using temporal information is an advantage, in other words, under what scenarios the temporal information is not available?
3) the proposed method has the assumption that the dynamics still follow the original distribution, but if the system is highly dynamic, is the proposed method still effective? Since the proposed method uses the continuity equation meaning the system is balanced.
4) the related work presented in the introduction is not logically sound to point out the contribution of the proposed method since the related work has some conditions meanwhile this paper does not fully clarify the difference.
5) the authors could use some figures to illustrate the high-level idea of the proposed method to increase the readability.

**Strengths:**

1) The paper presents the technical details in rigorous math formulation.
2) Broad empirical coverage by different categories of datasets.

**Weaknesses:**

1) readability: the paper is not sufficiently clear for non-experts, especially regarding the practical motivation and potential applications of the proposed framework.
2) justification of contribution: the paper does not adequately compare the proposed method with prior approaches mentioned in the Introduction.

**Questions:**

See detail comments in Summary.

---

> ### Author Response · Authors · 2025-11-17
>
> We thank the reviewer for providing helpful questions and suggestions, especially regarding the equilibrium hypothesis and improving readability.
>
> 1.  **Regarding temporal information:** Inferring potential dynamics from a data distribution lacking explicit temporal information is an important problem in various domains. As we mentioned in the introduction, much of the data in cosmology fundamentally consists of *snapshots*, since the evolutionary timescales of planets or stars are vastly longer than human history. For instance, when studying the evolution of a main-sequence star, one cannot effectively track a single star's trajectory because it changes too slowly. Instead, researchers use a collection of data from many stars to infer the potential dynamics. This is exactly what the **Hertzsprung-Russell diagram** achieves: by plotting the temperature and luminosity of numerous stars, clear patterns emerge that allow us to understand stellar evolution. Furthermore, our method does not exclude the integration of additional physical, biological, or other domain-specific information.
>
> 2.  **Regarding the equilibrium hypothesis:** This is an insightful observation and precisely why we term our method "equilibrium flow." We **hypothesize** that the observed system is in an equilibrium state; that is, its overall distribution does not change over time (or changes slowly compare to microscopic dynamics). While we acknowledge that not all systems are in equilibrium, many systems do exhibit this feature. In biology, for example, processes like blood flow and neural information transmission are fast, while macroscopic features (e.g., an organism's morphology or long-term memory) are relatively slow to change. This suggests that many systems that have "fast microscopic dynamics" and "slow macroscopic dynamics" are suitable for our hypothesis and method.
>
>     Regarding the term "highly dynamic" mentioned by the reviewer, we would welcome clarification on its precise definition in this context. However, if we assume "highly dynamic" refers to chaotic systems, such as the **Lorenz system**, our point still holds. For such systems, it is well-known that they possess a chaotic attractor, which itself has a stable, time-invariant statistical distribution. For non-equilibrium states, we acknowledge our method is not suitable. However, the applicability of our equilibrium assumption can often be assessed before applying the method.
>
> 3.  **Regarding the writing:** We will revise the paper to improve its clarity and will add high-level illustrations in the next version.
>
> 4.  **Regarding advantages over existing methods:** A direct comparison to many existing methods is not straightforward, as our approach is fundamentally different. Most existing methods require temporal data, whereas our method operates without any explicit time information. This fundamental difference makes a direct "apples-to-apples" comparison difficult. However, as discussed, inferring dynamics with absolutely no time information has critical applications in its own right.
>
>     Other methods that also do not require time information often rely on a "pseudo-time" variable. This solution is particularly problematic for systems with circular or periodic behaviors. If $T$ is the periodic time, any time $t$ and $t + kT$ (where $k \in \mathbb{Z}$) would represent the same state, making a simple pseudo-time mapping ambiguous. This ambiguity makes it difficult for such methods to model periodic dynamics correctly. In contrast, our method removes explicit time entirely and uses an ODE to describe the flow. As we show in Figure 2, our method can successfully capture various forms of periodic dynamics.

---

> > ### Comment · Reviewer_8dfP · 2025-11-27
> >
> > Thank you to the authors for the detailed response. I have read it carefully.
> >
> > The response goes a good way toward addressing several of my original concerns, in particular by (i) clarifying the motivation for working purely with snapshot data and giving concrete scientific scenarios where temporal information is unavailable or impractical, and (ii) explaining more clearly the intended meaning and applicability of the equilibrium assumption (including for systems like the Lorenz attractor). I also appreciate the commitment to improve the exposition and add higher-level illustrations in a revised version.
> >
> > I still think the final version would benefit from a brief, more explicit discussion of when the proposed approach is and is not appropriate (e.g., clearly non-equilibrium systems), and from a slightly clearer positioning with respect to prior methods that require temporal information or pseudo-time.

---

### Meta-Review · Area_Chair_yxrY · 2026-01-03

**Summary:**

The reviewers agree that the problem of estimating a vector field from static snapshots of data is important (and very challenging), and the paper demonstrated interesting approaches in this direction. The reviewers also noted a primary concern of ill-posedness of the problem, and questioned whether or not the recovered vector field has captured the right dynamics, besides qualitatively reproducing the invariant distribution. The reviewers also noted some clarity issues and suggested some real-life experiments to demonstrate the approach.

**Reviewer Concerns:**

While minor technical clarifications are addressed, the core issue of ill-posedness has not been sufficiently addressed. It is clear that the invariant distribution only characterises the gradient part of the vector field, and hence *it does not sufficiently constrain the dynamics*. There is not sufficient evidence to demonstrate that any non-trivial dynamical features of the vector field (e.g. quantitative Lyapunov exponents) have been correctly recovered (nor should one expect so, in my opinion). I also agree with the reviewer that the paper may be over-claiming when it comes to "artificial life".

**Reviewer Scores:**

No major change in reviewer scores are expected.

---

### Decision · Program_Chairs · 2026-01-26

Reject